# WW Domain-Containing E3 Ubiquitin Protein Ligase 1 (WWP1) as a Factor in Obesity-Related Metabolic Disorders: Emerging Molecular Mechanisms in Metabolic Tissues

**DOI:** 10.3390/ijms26189172

**Published:** 2025-09-19

**Authors:** Yuka Nozaki, Yuhei Mizunoe, Masaki Kobayashi, Yoshikazu Higami

**Affiliations:** 1Faculty of Pharmaceutical Sciences, Tokyo University of Science, Tokyo 125-8585, Japan; 2Department of Nutrition and Food Science, Graduate School of Humanities and Sciences, Ochanomizu University, Tokyo 112-8610, Japan; 3Institute for Human Life Science, Ochanomizu University, Tokyo 112-8610, Japan; 4Division of Cell Fate Regulation, Research Institute for Biomedical Science, Tokyo University of Science, Chiba 278-1501, Japan

**Keywords:** WWP1, obesity, white adipose tissue, hepatic steatosis, oxidative stress, lipolysis, insulin signaling

## Abstract

WW domain-containing E3 ubiquitin protein ligase 1 (WWP1) is a member of the homologous to E6AP C-terminus-type E3 ubiquitin protein ligase family. Although WWP1 plays a role in several human diseases, including infectious diseases, neurological disorders, and cancers, there is emerging evidence that WWP1 is also associated with metabolic disorders. In this review, we discuss the regulation and molecular function of WWP1 and its contribution to obesity-related metabolic disorders, particularly in white adipose tissue and the liver. We highlight the need for further research to deepen our understanding of how WWP1 may be implicated in metabolic dysfunction and facilitate the development of novel therapeutic strategies that target WWP1.

## 1. Introduction

Obesity is defined as abnormal and/or excessive fat accumulation in the body and is commonly associated with other metabolic disorders. Obesity has become a serious problem for human health in the modern world: In 2022, the World Health Organization reported that more than 2.5 billion adults aged 19 years and older were overweight, of whom 890 million were obese [1]. Obesity is strongly associated with insulin resistance, with chronic inflammation and oxidative stress as major mechanisms underlying this relationship [2]. As a result, obesity increases the risk of a wide range of metabolic disorders, including type 2 diabetes (T2D), cardiovascular disease, chronic kidney disease, and non-alcoholic fatty liver disease (NAFLD, now referred to as metabolic dysfunction-associated steatotic liver disease [MASLD]) [2,3].

Adipose tissue is broadly classified into three types: white adipose tissue (WAT), which represents more than 95% of fat mass; brown adipose tissue (BAT), which represents 1–2% of fat mass; and beige (or brite) adipose tissue, which is interspersed within WAT and difficult to quantify. Among adipose tissue types, WAT is the body’s most effective lipid storage organ and plays an important role in systemic metabolic homeostasis. White adipocytes, which are the main component of WAT and contain unilocular lipid droplets, provide endogenous energy via lipolysis during periods of low nutritional availability [4]. Lipolysis is a catabolic pathway that releases fatty acids as fuel to peripheral tissues from triglyceride (TG) stores in adipocytes [5]. During the development of obesity, WAT becomes hypertrophic and stores excess lipid as TG; hypertrophy of mature adipocytes disrupts the balance of adipokine secretion and increases the release of pro-inflammatory cytokines that induce systemic insulin resistance, chronic inflammation, and oxidative stress [2,3,6]. In addition, obesity is associated with impaired catecholamine-stimulated lipolysis and decreased TG turnover, which may further contribute to the excess accumulation of body fat [7,8,9].

In people with overweight or obesity, ectopic fat deposition, including in the liver and skeletal muscle, is common; this can lead to insulin resistance [10] and contribute to cardiovascular disease [11]. Specifically, ectopic fat accumulation in the liver is known as NAFLD, which is the most common liver disease worldwide [12,13]. NAFLD is an obesity-associated risk factor for serious liver diseases, including cirrhosis and hepatocellular carcinoma [12], as well as T2D and cardiovascular disease [14]. In 2023, NAFLD was renamed MASLD as a subclassification of steatotic liver disease; this renaming aims to broaden diagnostic criteria and tailor treatment strategies for the disease [15].

WWP1 (WW domain-containing E3 ubiquitin protein ligase 1; also known as TIUL1 [TGIF-interacting ubiquitin ligase 1] or AIP5 [atrophin-1 interacting proteins]) is a homologous to E6AP C-terminus (HECT)-type E3 ubiquitin protein ligase [16]. HECT-type E3 ligases are categorized into three families based on the variability of their N-terminal domains: the neural precursor cell expressed developmentally downregulated protein (NEDD) 4 family, the HERC (HECT and RLD domain-containing) family, and other E3 ligases [17,18,19]. WWP1 belongs to the NEDD4 family along with NEDD4-1, NEDD4-2, WWP2, itchy E3 ubiquitin protein ligase (ITCH), NEDD4-like E3 ubiquitin protein ligase (NEDL) 1 and 2, and SMAD-specific E3 ubiquitin protein ligase (Smarf) 1 and 2 [20,21,22]. WWP1 has a C2 domain at its N-terminal, four WW domains in its central region, and a HECT domain at its C-terminal [16,23]. The WW domains bind to proline-rich sequences (the PY motif) of substrate proteins, while the C2 domain determines the subcellular localization of the molecule [23]. WWP1 is ubiquitously expressed in almost all human tissue, and has been implicated in cancer (e.g., prostate, gastric, and breast cancer), infectious diseases (e.g., the Ebola, Hepatitis B, and leukemia viruses and COVID-19), muscular dystrophy, neurological diseases (e.g., Troyer syndrome, hereditary spastic paraplegias, and dentatorubral–pallidoluysian atrophy), and aging; WWP1 is an important modulator of a wide variety of intracellular processes [24]. In recent years, the novel role of WWP1 as an obesity-associated molecule relevant to metabolic diseases has been reported. In this review, we present the latest findings on WWP1 as a potential drug target and its relevance in obesity-associated disorders and metabolic tissues, particularly WAT and the liver.

## 2. The Regulation of WWP1 Expression in Mammals

As mentioned above, the expression of WWP1 is upregulated in several diseases, particularly cancer: it promotes cancer progression and contributes to a poor prognosis in several human cancers [25]. Although WWP1’s exact regulation mechanisms are unclear, several candidates have been identified as WWP1 expression regulators. Transforming growth factor β (TGFβ) [26] and tumor necrosis factor α (TNFα) [27] stimulate transcription of the WWP1 gene via unknown mechanisms in cancer progression. In addition, the transcription factors myelocytomatosis (MYC) and p53 positively regulate the expression of WWP1 by directly binding to its promoter region [28,29,30]. In addition, several microRNAs negatively regulate WWP1 expression [31,32,33,34]. However, almost all reports on the expression mechanisms of WWP1 have used cancer models, and little is known about WWP1 regulation in metabolic disorders.

Previously, we demonstrated that *Wwp1* mRNA expression was upregulated in the obese Zucker fatty rat (fa/fa) compared with the Zucker lean rat (+/+) and that WWP1 mRNA and protein expression were increased in WAT derived from mice with high-fat diet (HFD)-induced obesity [35]; however, increases in *Wwp1* mRNA expression levels in response to an HFD were not observed in the WAT of *p53* knockout (KO) mice [35]. These results suggest that obesity increases WWP1 expression in a p53-dependent manner, specifically in WAT.

## 3. Lipid Homeostasis Regulation by WWP1 in Adipocytes

Catecholamines are hormones with pronounced lipolytic action. Obesity is associated with an increase in basal lipolysis [8,36] but a decrease in catecholamine-stimulated lipolysis [7]. The β3 adrenergic receptor (β3AR), which is encoded by the adrenoceptor β3 (*Adrb3*) gene, is suppressed in obese WAT, both in non-human mammals [37,38,39] and humans [8,40,41]. In addition, recent human cohort studies have revealed that *Adrb3* expression in subcutaneous WAT derived from women with or without obesity is negatively correlated with body mass index (BMI) [9]. These results suggest that *Adrb3* expression is closely linked with adiposity. In our previous study using *Wwp1* KO mice, a distinctive feature pathway enrichment analysis of a comprehensive RNA-seq transcriptomic dataset showed an increase in genes related to nervous system pathways in the WAT of obese *Wwp1* KO mice. Specifically, *Adrb3* mRNA expression was decreased after HFD feeding in wild-type (WT) mice, an effect that was completely abolished in *Wwp1* KO mice [42]. In another study, which used mice heterozygous for *Nedd4*, another member of the NEDD4 family of E3 ubiquitin ligases, primary adipocytes had higher β2AR protein levels than those from WT mice [43]. Although the precise mechanisms by which NEDD4 and WWP1 regulate AR expression in adipocytes are unclear, these findings suggest that β-AR-mediated lipolysis may be partially regulated by the NEDD4 family of E3 ubiquitin ligases.

Catecholamine-mediated lipolysis is initiated by the activation of β-ARs, which stimulate cyclic AMP (cAMP) production and activate protein kinase A. This, in turn, phosphorylates key targets such as hormone-sensitive lipase (HSL), adipose triglyceride lipase (ATGL), and perilipin, which promote the breakdown of stored TG [44,45]. In our study of *Wwp1* KO mice, although the amount of noradrenaline (NA), its end metabolite 3-methoxy-4-hydroxyphenylglycol (MHPG), and the ratio of MHPG to NA, which reflects NA metabolism, were not different, the phosphorylated HSL/HSL ratio was increased compared with that in WT mice [42]. These results suggest that WWP1, which is increased in obesity, may reduce catecholamine-stimulated lipolysis in adipocytes via suppression of *Adrb3* expression through unknown mechanisms, although these do not seem to involve neurotransmission or NA metabolism. Therefore, the inhibition of WWP1 expression combined with the administration of β3-AR agonists may become a novel treatment for catecholamine resistance in obesity and diabetes.

## 4. The Role of WWP1 in Protecting Against Golgi Apparatus Disruption and Oxidative Stress in Adipocytes

### 4.1. WWP1 Localizes to the Golgi Apparatus via Its C2 Domain and Protects Golgi Morphology

The Golgi apparatus is a membrane organelle at the center of the secretory pathway that ensures accurate protein glycosylation of surface and secreted proteins [46,47]. In addition, the synthesis of glycosaminoglycans such as chondroitin sulfate (CS) and heparan sulfate (HS), which attach to core proteins, occurs predominantly in the Golgi apparatus [48]. HS and CS glycosylation increases during adipocyte differentiation and promotes lipid accumulation in 3T3-L1 adipocytes [49]. A recent report argued that HS and CS proteoglycan levels are associated with T2D and obesity [50], but highlighted that the specific roles of these proteoglycans during metabolic disease, including obesity, remain unclear.

The Human Protein Atlas Project demonstrated that WWP1 localizes to the Golgi apparatus by immunofluorescence microscopy [51,52]. Consistent with this, we previously reported that exogenous WWP1 specifically localized to the Golgi apparatus, but not the endoplasmic reticulum or mitochondria, in 3T3-L1 adipocytes [53]. Notably, a WWP1 mutant lacking the C2 domain, which is responsible for subcellular localization [23], failed to localize to the Golgi, which suggests that the C2 domain is essential for WWP1’s Golgi localization [53]. Monensin rapidly disrupts Golgi morphology, increases Golgi size, and induces the Golgi stress response via the transcription factor E3 (TFE3) pathway [54,55,56]. We found that exogenous WWP1 prevented monensin-induced Golgi apparatus disruption and that these effects were similar but less robust with C2 domain-lacking WWP1 than with full-length WWP1 [53]. Furthermore, WWP1 depletion using short hairpin RNA (shRNA) in 3T3-L1 adipocytes decreased CS and HS [53]. Taken together, these results suggest that WWP1 may contribute to the synthesis of glycosaminoglycans by maintaining the structure of the Golgi apparatus.

### 4.2. WWP1 Modulates Oxidative Stress in WAT During Obesity

There is a close relationship between obesity and WAT oxidative stress [57]. Furukawa et al. reported that fat accumulation correlates with plasma lipid peroxidation, represented by thiobarbituric acid reactive substrate (TBARS) levels, in people with obesity. In addition, reactive oxygen species (ROS) accumulate in the WAT of nondiabetic obese KKAy model mice, and are accompanied by increased expression of nicotinamide adenine dinucleotide phosphate (NADPH) oxidase and decreased expression of antioxidative enzymes [58]. Moreover, in vitro, oxidative stress causes an increase in 3T3-L1 adipocyte and human adipose-derived stem cell differentiation [58,59,60]. WWP1 appears to have a role in ameliorating oxidative stress in adipocytes. For example, WWP1-overexpressing 3T3-L1 adipocytes exhibit a decline in paraquat and palmitic acid-induced increases in ROS levels represented by dichlorofluorescein diacetate (DCFDA) fluorescence. By contrast, depletion of WWP1 using shRNA knockdown elevates DCFDA fluorescence [35]. In addition, HFD-fed *Wwp1* KO mice have higher glutathione disulfide concentrations and a lower glutathione (GSH)/glutathione disulfide (GSSG) ratio in WAT than WT mice [61].

The mechanism by which WWP1 affects oxidative stress has not been fully elucidated; however, it may relate to its effects on Golgi apparatus stabilization. ROS are produced at low levels in healthy cells; however, when the production of free radicals or their products exceeds the capability of antioxidant defense mechanisms, ROS can contribute to the etiology of severe pathologies. In particular, ROS induces fragmentation and increases Golgi apparatus stress [62]. The Golgi apparatus is important for adipose tissue morphology and function. For example, a previous study supported the hypothesis that adiponectin is secreted following synthesis at the endoplasmic reticulum and processing in the Golgi/trans-Golgi network [63]. In addition, results from a recent study suggest that lipid phosphatidylinositol 4-phosphate (PtdIns4P), which is enriched in the Golgi apparatus, decreases the stability of ATGL and lipolysis during glucose depletion [64]. Another hypothesis regarding the anti-oxidative stress effects of WWP1 relates to lipotoxicity. We previously reported that compared with WT 3T3-L1 adipocytes, the saturated fatty acid palmitate induced mitochondrial ROS (represented by mitoSOX fluorescence) significantly less in *Wwp1* overexpressing cells, but significantly more in *Wwp1* knockdown cells [35]. As circulating levels of palmitate and other saturated free fatty acids are increased in obesity [65,66] and stimulate oxidative stress in obese adipose tissue [2], WWP1 may ameliorate cellular stress in obese WAT by suppressing lipotoxicity-induced ROS production and the associated Golgi stress (Figure 1).

## 5. WWP1 Decreases Insulin Sensitivity and Exacerbates Hepatic Steatosis

### 5.1. Systemic Depletion of WWP1 Improves Insulin Sensitivity in the Obese Liver

Insulin resistance is characterized by a reduction in the normal physiological function of insulin. Insulin resistance usually precedes pathologies such as T2D and metabolic syndrome, and is associated with conditions such as overweight and obesity [67,68]. In the hepatic insulin signaling pathway, insulin binds to the insulin receptor, which triggers the insulin receptor’s autophosphorylation and the subsequent phosphorylation of insulin receptor substrates. This activation initiates the phosphatidylinositol 3-kinase (PI3K) pathway, wherein phosphatidylinositol 4,5-bisphosphate (PI(4,5)P2) is converted to phosphatidylinositol 3,4,5-triphosphate (PI(3,4,5)P3), a key phospholipid second messenger. Phosphatase and tensin homolog (PTEN) negatively regulates this process by dephosphorylating PIP3 [69,70]. At the plasma membrane, accumulated PIP3 activates Akt [71], which, in the liver, suppresses hepatic glucose production and promotes lipogenesis by suppressing forkhead box protein O1 (Foxo1)-dependent activation of glucose-6-phosphatase and inhibiting glucokinase, respectively [72,73,74].

In the liver of HFD-induced obese *Wwp1* KO mice, intravenously administered insulin decreases PTEN expression and increases the pAkt/Akt ratio, which suggests that WWP1 may positively regulate PTEN. However, in other insulin-sensitive tissues, including WAT and skeletal muscle, insulin does not affect the pAkt/Akt ratio and PTEN levels. In addition, obese *Wwp1* KO mice have decreased hepatic TG contents [75]. Korenblat and colleagues suggested that decreased hepatic TG content is associated with enhanced insulin signaling in the liver [76]. Thus, WWP1 appears to impair hepatic insulin signal transduction via PTEN stabilization, at least in obese mice [75].

By contrast, WWP1 may negatively regulate PTEN in cancer [28,77]; indeed, WWP1 directly ubiquitinates PTEN and inhibits its dimerization and translocation to the cellular membrane in human prostate cancer [28]. Posttranslational modifications of PTEN, including phosphorylation, ubiquitination, sumoylation, and acetylation, can dynamically change its activity [78]. However, the regulation of PTEN cannot be explained by WWP1 alone, and tissue and/or status-specific regulatory mechanisms are also likely involved.

### 5.2. Systemic Depletion of WWP1 Improves Hepatic Fat Accumulation in Obese Mice

The hallmark feature of MASLD [15], an increase in intrahepatic TG content, occurs because of an imbalance between a complex interaction of metabolic events, with the presence of steatosis associated with adverse alterations in glucose, fatty acid, and lipoprotein metabolism [79,80]. We previously showed that *Wwp1* KO mice exhibited a lower increase in hepatic TG content and liver weight than that normally associated with an HFD [75]. Consistent with this finding, Chen and colleagues recently demonstrated that WWP1 targets enhancer-of-split and hairy-related protein 1 (SHARP1) for degradation by ubiquitinating its P-rich domain located in the C-terminus. Notably, point mutations within the P-rich domain of SHARP1 enhanced the inhibitory activity against MASLD compared with the WT protein. In fatty acid-treated HepG2 hepatocytes, the interaction between SHARP1 and CCAAT-enhancer binding protein beta (C/EBPβ) was weakened, which led to increased C/EBPβ binding to the *Wwp1* promoter and subsequent WWP1 upregulation. These results suggest that WWP1 expression is upregulated in the livers of MASLD model mice and humans through a positive feedback loop involving WWP1, SHARP1, and C/EBPβ [81]. However, while WWP1 expression increased in the liver of their MASLD model, it did not increase in the liver of our obese model mice, despite similar feeding periods. These differences may be due to variations between the high-fat, high-cholesterol feed (43.7% fat, 36.6% carbohydrate, 19.7% protein, and 0.203% cholesterol) used in their study and the HFD (32.0% crude lipid, 25.5% crude protein, and 2.9% crude fiber) used in ours. Further studies are required to identify the mechanism by which WWP1 is regulated in the obese liver.

### 5.3. Systemic Depletion of WWP1 Improves Systemic Insulin Sensitivity in Obese Mice

Recently, excessive insulin production (also called hyperinsulinemia) has been attracting attention because of its relation to obesity-induced metabolic disease. It is commonly accepted that hyperinsulinemia results from insulin resistance in glucose metabolism, which leads to hyperglycemia; this, in turn, stimulates pancreatic β-cells to release insulin to prevent more severe hyperglycemia [68,82]. In a previous study, HFD-induced obese *Wwp1* KO mice exhibited lower circulating insulin levels and improved insulin sensitivity compared with obese WT controls [61]. As WWP1 may impair hepatic insulin signaling [75], these findings suggest that the reduction in circulating insulin levels observed in obese *Wwp1* KO mice was likely caused by enhanced hepatic insulin signaling, which may contribute to improved systemic glucose metabolism.

## 6. Comparison of NEDD-Family-Deficient Mice and Obesity-Related Phenotypes

WWP1 is a member of the NEDD4 family of E3 ubiquitin ligases, which includes NEDD4-1, NEDD4-2, WWP2, ITCH, NEDL1/2, and Smurf1/2 [20,21,22]. These proteins perform a diverse range of cellular functions because of variations in their WW domain [83] and distinct substrate specificities [84]. Several family members, including WWP1, NEDD4, and ITCH, have been evaluated in the context of obesity using KO mice models. In particular, both heterogenous *Nedd4* KO (*Nedd4*^+/−^) [43] and *Itch* KO (Itch^−/−^) [85] mice display improved obese-related insulin resistance, as evaluated by the insulin tolerance test (ITT). Similarly, *Wwp1* KO mice exhibit enhanced insulin sensitivity, which suggests a shared role in modulating insulin resistance. Additionally, *Wwp1* KO and *Nedd4*^+/−^ mice both exhibit enhanced lipolysis, which is attributed to the upregulation of adrenergic receptor expression. By contrast, *Wwp1* KO mice and *Itch*^−/−^ mice exhibit a common phenotype of exacerbated hepatic steatosis (Table 1). Although direct comparisons among these mouse models are limited by differences in sex, feeding conditions, and age, these findings suggest that NEDD4 family proteins share a capacity to respond to metabolic stressors such as obesity and play key roles in regulating metabolic functions in energy-relevant tissues.

## 7. Conclusions and Future Perspectives

In this review, we summarized the molecular mechanisms of WWP1 in obesity-related metabolic dysfunction, particularly in WAT and the liver. In WAT, WWP1 appears to play dual roles: first, obesity-induced, p53-dependent upregulation of WWP1 suppresses catecholamine-stimulated lipolysis through downregulation of *Adrb3* expression; and second, WWP1 contributes to protection against intracellular stress, including oxidative and Golgi stress. Although the direction of regulation remains unclear, ROS regulates both lipolysis and lipogenesis [86]. Future studies should explore the detailed mechanisms underlying redox regulation and lipid homeostasis in adipocytes, particularly in relation to WWP1. In the liver, WWP1 suppresses insulin signaling and promotes the accumulation of TG, which exacerbates hepatic steatosis. The molecular mechanisms underlying the regulation of MASLD remain largely unknown; however, the quantitative regulation of WWP1 may be efficacious as a target for MASLD. In conclusion, targeting WWP1 may be useful in the treatment of obesity and related metabolic disorders through the dual functions of promoting lipolysis of WAT and insulin sensitivity in the liver (Figure 2).

Based on current knowledge, we anticipate that inhibition of WWP1 could represent a novel therapeutic approach not only for cancer, but also for metabolic diseases. However, several challenges remain in the development of WWP1-targeting therapeutic strategies. First, although WWP1 is ubiquitously expressed across various tissues, there is little evidence regarding tissue-specific differences in its expression; however, it is essential to target WWP1 in a tissue- and disease-specific manner to minimize off-target effects and enhance therapeutic precision. Second, although several WWP1 inhibitors have recently been developed, no studies have evaluated their effects in the context of metabolic diseases, and the precise inhibitory mechanisms of most compounds remain unclear. Therefore, further detailed investigation into WWP1-associated signaling pathways, as well as rigorous evaluation of the therapeutic efficacy of WWP1 inhibitors in metabolic disorders, is needed.

## Figures and Tables

**Figure 1 ijms-26-09172-f001:**
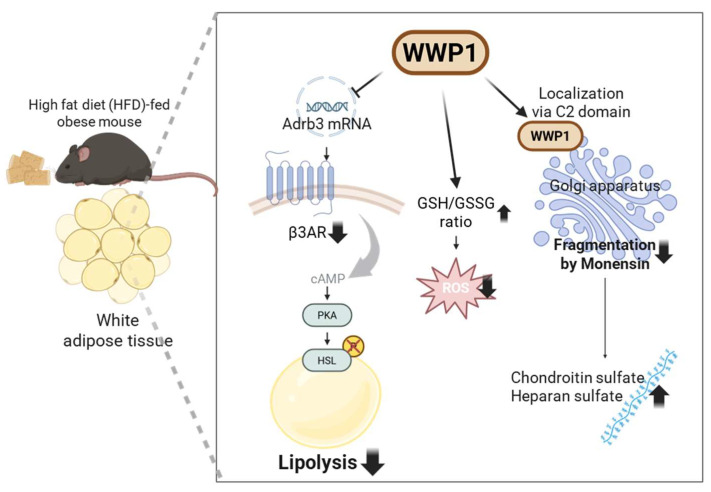
Physiological functions of WW domain-containing E3 ubiquitin protein ligase 1 (WWP1) in obese white adipose tissue (WAT). β3ARs, which are encoded by the *Adrb3* gene, promote lipolysis by increasing cAMP levels; this activates PKA, which subsequently phosphorylates HSL. WWP1 suppresses *Adrb3* mRNA expression and HSL phosphorylation. WWP1 also suppresses paraquat and palmitic acid-induced cellular ROS levels by regulating the GSH/GSSG ratio in WAT. Finally, WWP1 suppresses monensin-induced Golgi apparatus fragmentation and the Golgi stress response through localization to the Golgi apparatus via its N-terminal C2 domain. Therefore, WWP1 may function as a cellular stress regulator by mitigating ROS and the associated Golgi stress that are elevated in obese WAT. β3AR: beta 3-adrenergic receptor; *Adrb3*: adrenoceptor β3; cAMP: cyclic AMP; PKA: protein kinase A; HSL: hormone-sensitive lipase; ROS: reactive oxygen species; GSH: glutathione; GSSG: glutathione disulfide.

**Figure 2 ijms-26-09172-f002:**
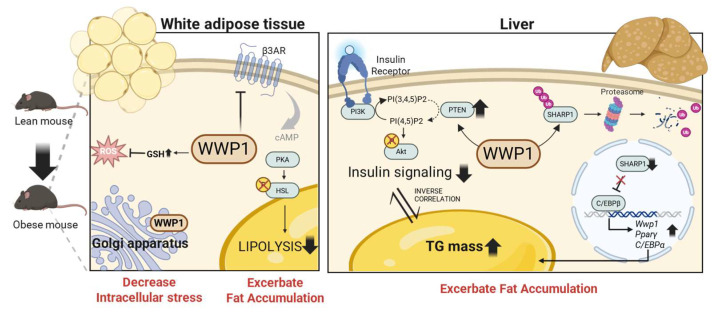
Diverse metabolic mechanisms of WW domain-containing E3 ubiquitin protein ligase 1 (WWP1) in obese white adipose tissue (WAT) and the liver. In WAT, WWP1 decreases intracellular stress by elevating the GSH/GSSG ratio and localizing to the Golgi apparatus via its C2 domain, which protects the Golgi structure. In the liver, WWP1 regulates PTEN expression and decreases Akt phosphorylation, thereby suppressing insulin signaling in obesity. Furthermore, WWP1 exacerbates TG accumulation, which may result from impaired insulin signaling. Moreover, SHARP1, which inhibits adipogenesis by regulating C/EBPβ, is ubiquitinated and degraded by WWP1, thereby contributing to the progression of MASLD through enhanced fat accumulation. GSH: glutathione; GSSG: glutathione disulfide; PTEN: phosphatase and tensin homolog; β3AR: beta 3-adrenergic receptor; Akt: Akt serine/threonine kinase; TG: triglyceride; SHARP1: enhancer-of-split and hairy-related protein 1; C/EBPβ: CCAAT/enhancer-binding protein β; MASLD: metabolic dysfunction-associated steatotic liver disease; ROS: reactive oxygen species; cAMP: cyclic AMP; PKA: protein kinase A; HSL: hormone-sensitive lipase; PI3K: phosphatidylinositol 3-kinase; PI(4,5)P2: phosphatidylinositol 4,5-bisphosphate; PI(3,4,5)P3: phosphatidylinositol 3,4,5-triphosphate; Ub: ubiquitin; PPARγ: peroxisome proliferator-activated receptor γ; C/EBPα: CCAAT/enhancer-binding protein α.

**Table 1 ijms-26-09172-t001:** Comparison of the phenotypes in metabolic tissues of obese NEDD-family depleted mice. GTT: glucose tolerance test; ITT: insulin tolerance test; Adrb3: beta 3-adrenergic receptor; HSL: hormone-sensitive lipase; WAT: white adipose tissue; PTEN: phosphatase and tensin homolog; IRβ: insulin receptor β; Akt: protein kinase B; B2AR: beta 2-adrenergic receptor; HE: hematoxylin and eosin.

Knockout Mice	Feed Composition and Diet	Parameter	Knockout Phenotypes	Notes	Reference
Wwp1 knockout (Wwp1^−/−^)	High fat diet (HFD32, CREA: 32.0% crude lipid, 25.5% crude protein, and 2.9% crude fiber)	HFD-fed for 8 weeks from 5-week-age of male C57BL/6J mice	Glucose tolerance (GTT and ITT)	↑ (Improved response to glucose and insulin)		[61]
Liver steatosis	↓ (Decreased in triglyceride contents)		[75]
HFD-fed for 10 weeks from 5-week-age of male C57BL/6J mice	Lipolysis	↑ (Increased Adrb3 expression and phosphorylation of HSL in WAT)	Insulin (1U/kg BW) i.p.	[42]
HFD-fed for 18 weeks from 5-week-age of male C57BL/6J mice	Insulin signaling	↑ (Increased phosphorylation of Akt and decreased PTEN with significant in liver, and non-significant in WAT and muscle)		[75]
Heterogenous Nedd4 knockout (Nedd4^+/−^)	High-fat, high-cholesterol diet (TD.06414 from Teklad, Harlan Laboratories: 43.7% fat, 36.6% carbohydrate, 19.7% protein, and 0.203% cholesterol)	HFHS-fed for 24 weeks from 6-week-age of male and female C57/BL6J mice	Glucose tolerance (GTT and ITT)	GTT has no change in both ND and HFD, ITT was impaired in HFD		[43]
Insulin signaling	↓ (Decreased in phosphorylation of IRβ and Akt in WAT, liver and muscle)	
Lipolysis	↑ (Increased serum glycerol)	16 weeks HFD feeding, After 10 mg/kg BW isoproterenol i.p.
↑ (Increased B2AR expression in WAT and primary adipocyte)	16 weeks HFD feeding
Liver steatosis	Not measured	
Itch knockout (Itch^−/−^)	High fat diet (GLP Mucedola Srl, Settimo Milanese: 32.0% crude lipid, 25.5% crude protein, and 2.9% crude fiber)	HFD-fed for 12 weeks from 6 to 8-week-age of male C57/BL10 mice	Glucose metabolism (GTT and HOMA-IR)	↑ (Improved response to glucose)		[85]
Insulin signaling	↑ (Increased in phosphorylation of Akt in muscle)	
Lipolysis	Not measured	
Liver steatosis	↓ (Decreased triglyceride contents)	
↓ (Decreased onset of steatosis in HE stain and lipid accumulation in Oil Red O stain)

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
