# Peer review of "WW Domain-Containing E3 Ubiquitin Protein Ligase 1 (WWP1) as a Factor in Obesity-Related Metabolic Disorders: Emerging Molecular Mechanisms in Metabolic Tissues"

_ijms, 2025, doi:10.3390/ijms26189172_

Round 1
Reviewer 1 Report
Comments and Suggestions for Authors
This review explores the role of WWP1, an E3 ubiquitin ligase, in obesity-related metabolic dysfunction, with a focus on its actions in white adipose tissue (WAT) and the liver. It suggests that targeting WWP1 could offer therapeutic benefits by promoting lipolysis in WAT and improving insulin sensitivity in the liver. The review lays a strong foundation for understanding the metabolic functions of WWP1 but highlights the need for further mechanistic and translational studies.
WWP1 is discussed in detail regarding its dual role in adipose tissue—suppressing lipolysis and offering cellular stress protection—and its involvement in hepatic insulin resistance and triglyceride accumulation. The review provides molecular insights, referencing key regulators such as p53, PTEN, Akt, SHARP1, and C/EBPβ, and positions WWP1 as a potential therapeutic target for obesity-related metabolic disorders.
However, the review does not clearly define the regulatory role of reactive oxygen species (ROS) in WWP1-mediated metabolism. It also notes that WWP1 exerts both beneficial and detrimental effects (e.g., cellular protection vs. insulin resistance), complicating its therapeutic targeting. The mechanisms involving WWP1 in MASLD remain insufficiently characterized, and much of the evidence is based on preclinical models, limiting the direct applicability to human conditions.
Please include the following recommendations for future research:
- Elucidate the interactions between reactive oxygen species (ROS), WWP1, and lipid metabolism.
- Investigate the potential of WWP1 inhibitors as therapeutic agents for metabolic disorders.
- Explore tissue-specific modulation of WWP1 to minimize off-target effects and enhance therapeutic precision.
In Table 1: Kindly delete the publication year from the table.
Author Response
We thank the reviewer for their helpful comments, which have been invaluable for improving our review manuscript. Please find our responses to the comments and suggested revisions that we received below.
This review explores the role of WWP1, an E3 ubiquitin ligase, in obesity-related metabolic dysfunction, with a focus on its actions in white adipose tissue (WAT) and the liver. It suggests that targeting WWP1 could offer therapeutic benefits by promoting lipolysis in WAT and improving insulin sensitivity in the liver. The review lays a strong foundation for understanding the metabolic functions of WWP1 but highlights the need for further mechanistic and translational studies.
WWP1 is discussed in detail regarding its dual role in adipose tissue—suppressing lipolysis and offering cellular stress protection—and its involvement in hepatic insulin resistance and triglyceride accumulation. The review provides molecular insights, referencing key regulators such as p53, PTEN, Akt, SHARP1, and C/EBPβ, and positions WWP1 as a potential therapeutic target for obesity-related metabolic disorders.
However, the review does not clearly define the regulatory role of reactive oxygen species (ROS) in WWP1-mediated metabolism. It also notes that WWP1 exerts both beneficial and detrimental effects (e.g., cellular protection vs. insulin resistance), complicating its therapeutic targeting. The mechanisms involving WWP1 in MASLD remain insufficiently characterized, and much of the evidence is based on preclinical models, limiting the direct applicability to human conditions.
Please include the following recommendations for future research:
Elucidate the interactions between reactive oxygen species (ROS), WWP1, and lipid metabolism.
We previously reported that palmitate-induced mitochondrial ROS, represented by mitoSOX fluorescence, was significantly reduced in WWP1-overexpressing 3T3-L1 adipocytes, but elevated in WWP1-knockdown 3T3-L1 adipocytes in response to palmitic acid (Kobayashi et al., 2019). Circulating levels of palmitate and other saturated fatty acids are elevated during obesity and increase mitochondrial ROS levels (Lambertucci et al., 2008; Yeop Han et al., 2010)]; therefore, our findings suggest that WWP1 may suppress lipotoxicity-induced mitochondrial ROS production.
To address the reviewer’s comment, we have revised the text based on the above description in section 4.2, “WWP1 modulates oxidative stress in WAT during obesity,” in the manuscript (page 5, line 189).
References:
Kobayashi, M., Hoshino, S., Abe, T., Okita, N., Tagawa, R., Nagai, W., Konno, R., Suzuki, Y., Furuya, K., Ishikawa, N., Okado, H., Oku, M., Iwamoto, M., Miura, Y., Sudo, Y., & Higami, Y. (2019). Identification of WWP1 as an obesity-associated E3 ubiquitin ligase with a protective role against oxidative stress in adipocytes. Biochem Biophys Res Commun, 508(1), 117-122. https://doi.org/10.1016/j.bbrc.2018.11.127
Lambertucci, R. H., Hirabara, S. M., Silveira Ldos, R., Levada-Pires, A. C., Curi, R., & Pithon-Curi, T. C. (2008). Palmitate increases superoxide production through mitochondrial electron transport chain and NADPH oxidase activity in skeletal muscle cells. Journal of Cellular Physiology, 216(3), 796-804. https://doi.org/10.1002/jcp.21463
Yeop Han, C., Kargi, A. Y., Omer, M., Chan, C. K., Wabitsch, M., O'Brien, K. D., Wight, T. N., & Chait, A. (2010). Differential effect of saturated and unsaturated free fatty acids on the generation of monocyte adhesion and chemotactic factors by adipocytes: dissociation of adipocyte hypertrophy from inflammation. Diabetes, 59(2), 386-396. https://doi.org/10.2337/db09-0925
Investigate the potential of WWP1 inhibitors as therapeutic agents for metabolic disorders.
Although WWP1 inhibitors have been developed, no studies have evaluated the effects of WWP1 inhibitors on metabolic diseases, and the inhibitory mechanisms of most WWP1 inhibitors remain unclear. Among inhibitors, only I3C blocks WWP1-mediated ubiquitination of PTEN, and thereby suppresses tumor formation driven by the PI3K–AKT pathway (Lee et al., 2019). By contrast, in the context of insulin signaling in the obese liver, WWP1 deficiency decreases PTEN expression and increases the pAKT/AKT ratio, which suggests that WWP1 may positively regulate PTEN in metabolic tissues (Nozaki et al., 2023). Therefore, the use of I3C in the context of metabolic diseases should be approached with caution, and further detailed investigation into the signaling pathways involving WWP1 is required, along with rigorous evaluation of the therapeutic efficacy of WWP1 inhibitors in metabolic diseases.
To address the reviewer’s comment, we have revised the text based on the above description in section 7, “Conclusion and future perspectives,” in the manuscript (page 9, line 313).
References:
Lee, Y. R., Chen, M., Lee, J. D., Zhang, J., Lin, S. Y., Fu, T. M., Chen, H., Ishikawa, T., Chiang, S. Y., Katon, J., Zhang, Y., Shulga, Y. V., Bester, A. C., Fung, J., Monteleone, E., Wan, L., Shen, C., Hsu, C. H., Papa, A.,…Pandolfi, P. P. (2019). Reactivation of PTEN tumor suppressor for cancer treatment through inhibition of a MYC-WWP1 inhibitory pathway. Science, 364(6441). https://doi.org/10.1126/science.aau0159
Nozaki, Y., Kobayashi, M., Wakasawa, H., Hoshino, S., Suwa, F., Ose, Y., Tagawa, R., & Higami, Y. (2023). Systemic depletion of WWP1 improves insulin sensitivity and lowers triglyceride content in the liver of obese mice. FEBS Open Bio, 13(6), 1086-1094. https://doi.org/10.1002/2211-5463.13610 (FEBS Open Bio)
Explore tissue-specific modulation of WWP1 to minimize off-target effects and enhance therapeutic precision.
Although WWP1 is ubiquitously expressed across various tissues, no detailed studies have addressed tissue-specific differences in its expression levels. As the reviewer mentions, further investigations should consider such tissue-dependent expression patterns of WWP1 to minimize off-target effects and enhance therapeutic precision.
To address the reviewer’s comment, we have revised the text based on the above description in section 7, “Conclusion and future perspectives,” in the manuscript (page 9, line 313).
In Table 1: Kindly delete the publication year from the table.
We have deleted the publication year from Table 1.
Reviewer 2 Report
Comments and Suggestions for Authors
Dear authers,
this is a well written review paper. I have only very few points to be revised as they are not clear this theme complex.
This is e.g. p.4, the part concerning Golgi apparatus and glycosaminoglycans (ll. 145-150). Please explain what this function has to do with obesity.
p.6, line 247-252: Please revise this chapter. It is not totally clear for me, why the Wwp+ KO have higher insulin sensitivity only due to reduced insulin serum levels? Please reformulate this chapter.
Conclusion:
The conclusion has more the character of an abstract. Please revise this part to the structure of a real conclusion.
Author Response
We thank the reviewer for their helpful comments, which have been invaluable for improving our review manuscript. Please find our responses to the comments and suggested revisions that we received below.
this is a well written review paper. I have only very few points to be revised as they are not clear this theme complex.
This is e.g. p.4, the part concerning Golgi apparatus and glycosaminoglycans (ll. 145-150). Please explain what this function has to do with obesity.
As discussed in our review, the synthesis of glycosaminoglycans (GAGs) such as chondroitin sulfate (CS) and heparan sulfate (HS), which attach to core proteins, occurs predominantly in the Golgi apparatus. In addition, WWP1 depletion decreases the mass of both CS and HS in 3T3-L1 adipocytes. Pessentheiner and colleagues reported in their 2020 review article that HS and CS proteoglycan levels are associated with type 2 diabetes and obesity (Pessentheiner et al., 2020). However, they argued that the specific roles of these proteoglycans in metabolic disease, including obesity, remain unclear.
To address the reviewer’s comment, we have revised the text based on the above description in section 4.1, “WWP1 localizes to the Golgi apparatus via its C2 domain and protects Golgi morphology,” in the manuscript (page 4, line 139).
Reference:
Pessentheiner, A. R., Ducasa, G. M., & Gordts, P. (2020). Proteoglycans in Obesity-Associated Metabolic Dysfunction and Meta-Inflammation. Frontiers in Immunology, 11, 769. https://doi.org/10.3389/fimmu.2020.00769
p.6, line 247-252: Please revise this chapter. It is not totally clear for me, why the Wwp+ KO have higher insulin sensitivity only due to reduced insulin serum levels? Please reformulate this chapter.
We appreciate your constructive comments to improve this review. As you point out, the original text lacked a sufficient description of the relationship between serum insulin levels and improved insulin sensitivity in Wwp1 KO mice. In a previous study, HFD-induced obese Wwp1 KO mice exhibited lower circulating insulin levels compared with obese wild-type controls. Consistent with this, glucose and insulin tolerance tests demonstrated improved insulin sensitivity and glucose tolerance in obese Wwp1 KO mice. Furthermore, these mice had reduced hepatic triglyceride content. Korenblat and colleagues have suggested that decreased hepatic TG content is associated with enhanced hepatic insulin signaling (Korenblat et al., 2008). Therefore, these findings suggest that the reduction in circulating insulin levels in obese Wwp1 KO mice was likely attributable to improved hepatic insulin responsiveness, which may contribute to enhanced systemic glucose metabolism.
To address the reviewer’s comment, we have revised the manuscript by adding a description of the relationship between hepatic TG levels and hepatic insulin sensitivity to section 5.1, “Systemic depletion of WWP1 improves insulin sensitivity in the obese liver” (page 6, lines 229), and the relationship between serum insulin levels and systemic insulin sensitivity in obese Wwp1 KO mice has been clarified in section 5.3, “Systemic depletion of WWP1 improves systemic insulin sensitivity in obese mice” (page 7, lines 266).
Reference:
Korenblat, K. M., Fabbrini, E., Mohammed, B. S., & Klein, S. (2008). Liver, muscle, and adipose tissue insulin action is directly related to intrahepatic triglyceride content in obese subjects. Gastroenterology, 134(5), 1369-1375. https://doi.org/10.1053/j.gastro.2008.01.075
Conclusion:
The conclusion has more the character of an abstract. Please revise this part to the structure of a real conclusion.
To address the reviewer’s comment, the Conclusion section has been revised and retitled “Conclusion and future perspectives.” To this section, we have added a concise summary of key findings, discussed current limitations, and included future research directions (page 9, lines 313).
Reviewer 3 Report
Comments and Suggestions for Authors
This interesting narrative review presents data about the role of WW domain-containing E3 ubiquitin protein ligase 1 (WWP1) in obesity-related metabolic disease, with particular focus on white adipose tissue (WAT) and the liver. By examining the available literature, it becomes clear that the data on this subject remain quite limited, with Nozaki et al.—the authors of the current manuscript—having contributed substantially to this field. Their research is important and worthy of recognition. The topic is of interest both at the basic science level and for its potential clinical implications in the future. However, there are several aspects of the manuscript that could be improved in order to enhance its scientific clarity, accuracy, and overall quality. Most of the necessary revisions concern language, structure, and presentation. Below, I outline my specific comments and suggestions:
1. Introduction
- Begin with the definition of obesity, followed by epidemiological data, and conclude with its associated comorbidities.
- Lines 32–35: This section should be revised. It should be emphasized that obesity is strongly associated with insulin resistance (IR), with inflammation and oxidative stress representing major underlying mechanisms. However, not all conditions listed are strictly “metabolic-related diseases” (e.g., cancer). A clearer presentation of this distinction would improve accuracy.
- Since lines 48–58 discuss the replacement of the term NAFLD with MASLD, it is recommended that the term MASLD be used as early as line 33, with an explanation of the updated terminology (lines 48-53).
- A brief presentation of the types of adipose tissue would improve the logical flow and provide a smoother transition to the discussion of WAT in this review.
- Line 46: Please revise to: “In people with overweight or obesity” (the preferred terminology is “people with obesity” rather than “obese people”). Please revise consistently throughout the text (e.g., lines 154 and 155).
2. Obesity-induced WWP1 expression in adipose tissue
Although the title suggests a detailed discussion of adipose tissue, the section is relatively short and begins with an introduction about the role of WWP1 in cancer, followed by only one adipose-related study. The title should be reconsidered to better reflect the content.
3. Lipid homeostasis regulation by WWP1 in adipocytes
- Line 101: Please clarify what is meant by the phrase “in non-human animals
4. The role of WWP1 in protecting against Golgi apparatus disruption and oxidative stress in adipocytes
- Lines 133–139: Revise for clarity and avoid redundancy.
- The description of Golgi apparatus characteristics (lines 143–149) should precede the discussion of WWP1’s role.
- Please include an abbreviations list in Figure 1 for clarity.
General Comments
1. Abbreviations are largely omitted in the text. They should be systematically included. For example, terms such as World Health Organization, type 2 diabetes, cardiovascular disease, and chronic kidney disease should be abbreviated after first mention. Moreover, the abbreviations list should be revised to ensure consistency.
2. The citation style within the text and in the reference list does not conform to the journal’s guidelines.
3. The manuscript would benefit from a careful revision to improve originality and avoid overlap in phrasing with primary research articles. Additionally, improvements in English grammar and style are needed.
4. Figure 2 and its legend seem misplaced in the conclusion section. They should be incorporated into the appropriate section of the manuscript. The conclusion should instead succinctly summarize key findings, limitations, and future perspectives.
Comments on the Quality of English Language
Improvements in English grammar and style are needed.
Author Response
We thank the reviewer for their thorough and helpful comments, which have been invaluable for improving our review manuscript. Please find our responses to the comments and suggested revisions that we received below.
This interesting narrative review presents data about the role of WW domain-containing E3 ubiquitin protein ligase 1 (WWP1) in obesity-related metabolic disease, with particular focus on white adipose tissue (WAT) and the liver. By examining the available literature, it becomes clear that the data on this subject remain quite limited, with Nozaki et al.—the authors of the current manuscript—having contributed substantially to this field. Their research is important and worthy of recognition. The topic is of interest both at the basic science level and for its potential clinical implications in the future. However, there are several aspects of the manuscript that could be improved in order to enhance its scientific clarity, accuracy, and overall quality. Most of the necessary revisions concern language, structure, and presentation. Below, I outline my specific comments and suggestions:
- Introduction
- Begin with the definition of obesity, followed by epidemiological data, and conclude with its associated comorbidities.
On the basis of the reviewer’s comment, we have added a definition of obesity in the first line of the Introduction section in the revised manuscript (page 1, line 28).
- Lines 32–35: This section should be revised. It should be emphasized that obesity is strongly associated with insulin resistance (IR), with inflammation and oxidative stress representing major underlying mechanisms. However, not all conditions listed are strictly “metabolic-related diseases” (e.g., cancer). A clearer presentation of this distinction would improve accuracy.
We appreciate your important suggestion. As you point out, the original text lacked a sufficient description of the relationship between obesity-associated diseases and insulin resistance. We have revised the text in the manuscript (page 1, line 32).
- Since lines 48–58 discuss the replacement of the term NAFLD with MASLD, it is recommended that the term MASLD be used as early as line 33, with an explanation of the updated terminology (lines 48-53).
We have added a clarification to the first mention of NAFLD, indicating that it is “now referred to as metabolic dysfunction-associated steatotic liver disease (MASLD)” in the manuscript (page 1, line 36).
- A brief presentation of the types of adipose tissue would improve the logical flow and provide a smoother transition to the discussion of WAT in this review.
We have added the following sentence before the discussion of WAT (page 1, line 38): “Adipose tissue is broadly classified into three types: white adipose tissue (WAT), which represents more than 95% of fat mass; brown adipose tissue (BAT), which represent 1%–2% of fat; and beige (or brite) adipose tissue, which is interspersed within WAT and difficult to quantify.”
- Line 46: Please revise to: “In people with overweight or obesity” (the preferred terminology is “people with obesity” rather than “obese people”). Please revise consistently throughout the text (e.g., lines 154 and 155).
We have revised the text as “in people with overweight or obesity” throughout the manuscript (page 2, line 53 and page 4, line 165).
- Obesity-induced WWP1 expression in adipose tissue
Although the title suggests a detailed discussion of adipose tissue, the section is relatively short and begins with an introduction about the role of WWP1 in cancer, followed by only one adipose-related study. The title should be reconsidered to better reflect the content.
We have changed the title to “The regulation of WWP1 expression in mammals” In line 83.
- Lipid homeostasis regulation by WWP1 in adipocytes
- Line 101: Please clarify what is meant by the phrase “in non-human animals
We have changed the word to “non-human mammals” from “non-human animals” In line 107.
- The role of WWP1 in protecting against Golgi apparatus disruption and oxidative stress in adipocytes
- Lines 133–139: Revise for clarity and avoid redundancy.
We have revised the text to improve clarity and avoid redundancy regarding the localization of WWP1 to the Golgi via its C2 domain in the manuscript (page 4, line 139).
- The description of Golgi apparatus characteristics (lines 143–149) should precede the discussion of WWP1’s role.
We have updated the description of Golgi apparatus characteristics before discussing WWP1’s role in the 4th section of the manuscript (page 4, line 139).
- Please include an abbreviations list in Figure 1 for clarity.
We have added an abbreviations list and slightly modified the legends for Figures 1 (page 5, line 200) and 2 (page 9, line 334s) to ensure consistency in notation.
General Comments
- Abbreviations are largely omitted in the text. They should be systematically included. For example, terms such as World Health Organization, type 2 diabetes, cardiovascular disease, and chronic kidney disease should be abbreviated after first mention. Moreover, the abbreviations list should be revised to ensure consistency.
We have removed the abbreviations list and instead included all abbreviations in parentheses at the first mention of each term throughout the manuscript. This approach has been applied consistently, except for the legends of Figures 1 and 2.
- The citation style within the text and in the reference list does not conform to the journal’s guidelines.
We have revised the citation style in both the main text and the reference list according to the formatting guidelines of IJMS (ACS style).
- The manuscript would benefit from a careful revision to improve originality and avoid overlap in phrasing with primary research articles. Additionally, improvements in English grammar and style are needed.
We have carefully revised the manuscript to improve originality and rephrased sentences that overlapped with previously published primary research articles. The original version of the manuscript submitted for first review had already undergone professional English editing by Edanz. Furthermore, the additional text added in response to the current review has also been professionally edited by Edanz. If you find any errors, we would appreciate it if you could kindly point them out.
- Figure 2 and its legend seem misplaced in the conclusion section. They should be incorporated into the appropriate section of the manuscript. The conclusion should instead succinctly summarize key findings, limitations, and future perspectives.
We have relocated Figure 2 and its legend to the appropriate section of the manuscript. In addition, the Conclusion section has been revised and retitled as “Conclusion and future perspectives.” To this, we have added a concise summary of key findings, discussed current limitations, and included future research directions.
Comments on the Quality of English Language
Improvements in English grammar and style are needed.
The original version of the manuscript submitted for first review had already undergone professional English editing by Edanz. Furthermore, the additional text added in response to the current review has also been professionally edited by Edanz. If you find any errors, we would appreciate it if you could kindly point them out.
Round 2
Reviewer 3 Report
Comments and Suggestions for Authors
The manuscript has been significantly improved. I recommend its publication.